# Pro-Inflammatory Effects of NX-3 Toxin Are Comparable to Deoxynivalenol and not Modulated by the Co-Occurring Pro-Oxidant Aurofusarin

**DOI:** 10.3390/microorganisms8040603

**Published:** 2020-04-21

**Authors:** Lydia Woelflingseder, Nadia Gruber, Gerhard Adam, Doris Marko

**Affiliations:** 1Department of Food Chemistry and Toxicology, Faculty of Chemistry, University of Vienna, 1090 Vienna, Austria; lydia.woelflingseder@univie.ac.at (L.W.); nadia.gruber@gmx.at (N.G.); 2Institute of Microbial Genetics, Department of Applied Genetics and Cell Biology (DAGZ), University of Natural Resources and Life Sciences, Vienna (BOKU), 3430 Tulln, Austria; gerhard.adam@boku.ac.at

**Keywords:** mycotoxin, trichothecene, NF-κB, intestinal inflammation, combinatory effects, food safety

## Abstract

The type A trichothecene NX-3, produced by certain *Fusarium graminearum* strains, is similar to the mycotoxin deoxynivalenol (DON), with the exception that it lacks the carbonyl moiety at the C-8 position. NX-3 inhibits protein biosynthesis and induces cytotoxicity to a similar extent as DON, but so far, immunomodulatory effects have not been assessed. In the present study, we investigated the impact of NX-3 on the activity of the nuclear factor kappa B (NF-κB) signaling pathway in direct comparison to DON. Under pro-inflammatory conditions (IL-1β treatment), the impact on cytokine mRNA levels of NF-κB downstream genes was studied in human colon cell lines, comparing noncancer (HCEC-1CT) and cancer cells (HT-29). In addition, potential combinatory effects with the co-occurring *Fusarium* secondary metabolite aurofusarin (AURO), a dimeric naphthoquinone known to induce oxidative stress, were investigated. NX-3 and DON (1 μM, 20 h) significantly activated a NF-κB regulated reporter gene to a similar extent. Both trichothecenes also enhanced transcript levels of the known NF-κB-dependent pro-inflammatory cytokines IL-8, IL-6, TNF-α and IL-1β. Comparing the colon cancer HT-29 and noncancer HCEC-1CT cells, significant differences in cytokine signaling were identified. In contrast, AURO did not affect NF-κB pathway activity and respective cytokine expression levels at the tested concentration. Despite its pro-oxidant potency, the combination with AURO did not significantly affect the immunomodulatory effects of the tested trichothecenes. Taken together, the present study reveals comparable potency of DON and NX-3 with respect to immunomodulatory and pro-inflammatory potential. Consequently, not only DON but also NX-3 should be considered as factors contributing to intestinal inflammatory processes.

## 1. Introduction

Mycotoxins are toxic, secondary metabolites produced by certain filamentous fungi, mainly belonging to the genera of *Fusarium, Aspergillus, Penicillium* and *Alternaria.* Contaminating food and feed pre- and postharvest, mycotoxins pose a potential risk to food safety and thus to human and animal health. After ingestion, the intestinal tract represents the first barrier of the host against food contaminants such as mycotoxins and is therefore the first line for many defense mechanisms. In order to protect the entire organism from the entrance of unwanted solutes, microorganisms and luminal antigens, a proper function of the intestinal barrier and its innate immune response is crucial. As a defense mechanism against external stressors, intestinal epithelial cells are able to secrete cytokines and chemokines, including transforming growth factor-α (TGF-α), interleukin-1 (IL-1), interleukin-6 (IL-6), interleukin-8 (IL-8) or interleukin-10 (IL-10) in order to activate the immune response and to recruit respective immune cells [1]. However, if not controlled properly, excessive immune response and cytokine release can lead to chronic intestinal inflammation and contribute to the progression of inflammatory disorders, such as inflammatory bowel diseases (IBDs) [2].

One of the most prevalent mycotoxins in temperate climate regions is the trichothecene mycotoxin deoxynivalenol (DON, vomitoxin, Figure 1A), which frequently contaminates grain- and cereal-based products [3,4,5]. Human biomonitoring studies revealed that, due to its ubiquitous occurrence, consumers are chronically exposed to low levels of DON [6,7,8]. The epoxide moiety at position C12-C13 is considered a key factor in the toxin’s main mechanism of action, the inhibition of eukaryotic protein synthesis. By binding to the 60S ribosomal subunit, DON causes a ribotoxic stress response [9,10], resulting in the activation of mitogen-activated protein kinases (MAPKs) and furthermore in the induction of apoptosis and inflammation [11]. Depending on dose, exposure frequency and duration, DON can induce both immunostimulatory and immunosuppressive effects [11,12,13]. Low dose exposure caused transcriptionally and post-transcriptionally upregulation of immunostimulating cytokines and chemokines, whereas high dose exposure was shown to promote apoptosis with concomitant immunosuppressive effects [11,13]. Modulatory effects of DON on cytokine production in intestinal tissue and intestinal epithelial cancer cell lines have been studied and reviewed extensively during the last decades [11,14,15]. DON was reported to induce the secretion of IL-8 via a nuclear factor “kappa-light-chain-enhancer of activated B cells”-mediated (NF-κB) mechanism in various human intestinal epithelial cells such as Caco-2 or HT-29 [16,17,18]. When mimicking an inflamed intestinal epithelium, co-exposure experiments with various pro-inflammatory stimuli including interleukin-1β (IL-1β), the tumor necrosis factor-α (TNF-α) or lipopolysaccharide (LPS) resulted in additive and synergistic effects regarding cytokine secretion and NF-κB activation [16]. However, the effects of DON on the inflammatory response have so far only been reported in cancer cell models, but not in noncancer intestinal epithelial cells.

Moreover, no toxicological characterization of the recently discovered type A trichothecene NX-3 (Figure 1B) regarding immunomodulatory effects has been performed to date. Structurally identified by Varga et al. [19] as an analogue of DON lacking the carbonyl moiety at the C8-position, this toxin was found to be produced by *F. graminearum* [20,21]. In vitro translation assays showed a similar inhibitory potency on protein biosynthesis as DON [19]. Likewise, cytotoxicity studies revealed comparable effects of both trichothecenes in human liver cancer (HepG2), nontransformed colon cells (HCEC-1CT) and colon cancer cells (HT-29) [22,23]. However, potential inflammatory properties of NX-3 have not been investigated yet.

Among many other aspects, oxidative stress plays an essential role in the pathogenesis and progression of IBDs [24]. Another *Fusarium* secondary metabolite reported to induce oxidative stress in human colon adenocarcinoma cells HT-29, is the coloring pigment aurofusarin (AURO, Figure 1C) [25]. Even though AURO was already isolated in 1937 from the mycelium of *Fusarium culmorum* [26], toxicological studies are still limited. In HT-29 and HCEC-1CT cells, Jarolim, Wolters, Woelflingseder, Pahlke, Beisl, Puntscher, Braun, Sulyok, Warth and Marko [25] reported AURO to be cytotoxic at concentrations ≥ 1 µM. Although AURO is frequently found at high concentrations in various food commodities [27,28,29], immunomodulatory effects have not been assessed yet. As *Fusarium* fungi can produce several mycotoxins simultaneously, infested food and feed may be contaminated by a high number of toxins. In recent years, human biomonitoring studies confirmed that humans are exposed to a variety of toxins [30,31,32]. Thus, the assessment of potential interactions of mycotoxins is crucial for proper risk assessment.

In the present study, we addressed the question whether the *Fusarium* metabolites NX-3 and AURO affect the NF-κB signaling pathway as individual compounds and as binary mixtures. Furthermore, we characterized alterations in the gene expression profiles of the pro-inflammatory cytokines IL-8, IL-6, TNF-α and IL-1β in a cancer and a noncancer colon cell line, mimicking an inflamed intestinal epithelium by IL-1β stimulation.

## 2. Materials and Methods

### 2.1. Chemicals and Reagents

DON was purchased from Romer Labs (Tulln, Austria). NX-3 was produced and purified by preparative HPLC from NX-2 (purity >99% according to LC–UV at 200 nm) as published by Varga, Wiesenberger, Hametner, Ward, Dong, Schofbeck, McCormick, Broz, Stuckler, Schuhmacher, Krska, Kistler, Berthiller and Adam [19]. AURO was purchased from Biovitica (purity: 97.5%; Biovitica Naturstoffe GmbH, Dransfeld, Germany). DON and NX-3 were dissolved in water (LC–MS grade) to obtain stock solutions of 10 mM, which were further dissolved, aliquoted and stored at −20 °C. AURO was dissolved in dimethyl sulfoxide (DMSO). Stock solutions of 1 mM were ultrasonicated for 5 min, aliquoted and stored at −80 °C. For each incubation, a new aliquot was thawed.

### 2.2. Cell Culture and Treatment

The human monocytic cell line THP1-Lucia™ NF-κB, deriving from the human THP-1 monocyte cell line by stable integration of an NF-κB-inducible luciferase reporter construct, was purchased from Invivogen (San Diego, CA, USA) and HT-29, a human colorectal adenocarcinoma cell line from the German Collection of Microorganisms and Cell Cultures (DSMZ, Braunschweig, Germany). THP1-Lucia™ were cultured in RPMI 1640 medium, HT-29 in Dulbecco’s Modified Eagle’s Medium (DMEM), both supplemented with 10% (*v/v*) heat-inactivated fetal bovine serum and 1% (*v/v*) penicillin–streptomycin (100 U/mL). THP1-Lucia™ were treated alternately with zeocin and normocin (100 µg/mL; Invivogen, San Diego, CA, USA). Noncancer human colon epithelial cells HCEC-1CT [33,34] were kindly provided by Prof. Jerry W. Shay (UT Southwestern Medical Center, Dallas, TX, USA). HCEC-1CT cells were cultivated in high glucose DMEM. Basal medium was supplemented with the following components: Medium 199 (10×; 2% (*v/v*)), HyClone™ Cosmic Calf™ Serum (2% (*v/v*)), gentamicin (50 μg/mL), 4-(2-hydroxyethyl)-1-piperazineethanesulfonic acid (20 mM), insulin-transferrin-selenium-G (10 μg/mL; 5.5 μg/mL; 6.7 ng/mL), hydrocortisone (1 μg/mL) and recombinant human epidermal growth factor (18.7 ng/mL). All three cell lines were subcultured every 3–4 d, maintained in humidified incubators at 37 °C and 5% CO_2_ and routinely tested for the absence of mycoplasm contamination. Cell culture media, supplements and material were purchased from GIBCO Invitrogen (Karlsruhe, Germany), Sigma-Aldrich (Munich, Germany) and Sarstedt AG & Co. (Nuembrecht, Germany). DON, NX-3, AURO and their combinations were added to the incubation solutions, resulting in a final solvent concentration of 1% (*v/v*) DMSO. In order to ensure data comparability, combinatory effects were always assessed on the same culture plate, in parallel to the individual substance.

### 2.3. NF-κB Reporter Gene Assay

DON, NX-3, AURO and their combinations were prepared in reaction tubes to be further diluted 1:100 by the addition of the THP1-Lucia™ cell suspension. Cells were counted, centrifuged for 5 min at 250 × *g* and resuspended at 1 × 10^6^ cells/mL in fresh, prewarmed growth medium. Cell suspension was added to the respective toxin preparations, mixed gently and transferred into a 96-well plate (100 µL/well). After 2 h of incubation at 37 °C and 5% CO_2_, cells were treated with LPS (10 ng/mL) and incubated for an additional 18 h. 1% (*v/v*) DMSO with and without LPS treatment were used as solvent control. Heat-killed *Listeria monocytogenes* (HKLM; 20 × 10^6^ cells/well) were used as positive control for Toll-like receptor-mediated activation of the NF-κB pathway. Following treatment, cells were centrifuged (250 × *g*, 5 min) and 10 µL of each supernatant were collected and the reporter gene assay was performed according to the manufacturer’s protocol. For luminescence measurements QUANTI-Luc™ (Invivogen, San Diego, CA, USA), containing the luciferase substrate coelenterazine, was used.

In parallel cellular metabolic activity was monitored by the alamarBlue^®^ assay (Invitrogen, Carlsbad, CA, USA). After supernatant collection for luciferase activity measurements, 10 µL of alamarBlue^®^ reagent (Invitrogen, Carlsbad, CA, USA) were added to the cells and incubated for 2 h. Subsequently, 50 µL/well were transferred to a black 96-well plate and fluorescence intensity was measured at 530/560 nm (excitation/emission). Both luciferase activity and fluorescence intensity measurements were performed on a Synergy^TM^ H1 hybrid multimode reader (BioTek, Bad Friedrichshall, Germany), assessing at least five independent experiments in duplicates.

### 2.4. Quantitative Analysis of Cytokine Gene Transcription

Gene transcription levels of up to four pro-inflammatory cytokines in two colon cell lines (HT-29: TNF-α, IL-1β, IL-8; HCEC-1CT: TNF-α, IL-1β, IL-8, IL-6) were analyzed by quantitative real-time PCR (qRT-PCR). Cells were seeded in 12-well plates (HT-29: 150,000 cells/well; HCEC-1CT: 50,000 cells/well) and allowed to grow for 48 h. Cells were incubated for 5 h, consisting of 2 h preincubation with the test compounds (DON, NX-3, AURO and the respective combinations) followed by IL-1β co-treatment (25 ng/mL) for additional 3 h. Total RNA was extracted using Maxwell^®^ 16 Cell LEV Total RNA Purification Kits (Promega, Madison, WI, USA) and reversed transcribed into complementary DNA (cDNA) by QuantiTect^®^ Reverse Transcription Kit (Qiagen, Hilden, Germany) according to the manufacturer’s protocols. cDNA samples were amplified in duplicates in presence of gene specific primers (QuantiTect^®^ Primer Assays, Qiagen, Hilden, Germany) and QuantiTect^®^ SYBR Green Master Mix (Qiagen, Hilden, Germany) using a StepOnePlus™ System (Applied Biosystems, Foster City, CA, USA). The following primer assays were used: β-actin (ACTB1, Hs_ACTB1_1_SG, QT00095431); glyceraldehyde 3-phosphate dehydrogenase (GAPDH, Hs_GAPDH_1_SG, QT00079247); TNF-α (Hs_TNF_1_SG; QT00029162); IL-1β (Hs_IL1B_1_SG, QT00021385); IL-8 (Hs_CXCL8_1_SG; QT00000322); IL-6 (Hs_IL6_1_SG, QT00083720). The applied PCR protocol included 15 min enzyme activation at 95 °C, 45 cycles of 15 s at 94 °C, 30 s at 55 °C and 30 s at 72 °C. For the quantification of the fluorescence signal and further data analysis, StepOnePlus^®^ software (Applied Biosystems, Foster City, CA, USA) was used. For each tested sample, at least five independent experiments were performed. Presented transcript data were normalized to the mean of transcript levels of endogenous control genes (ACTB1, GAPDH) by applying the ΔΔCt-method [35] for relative quantification. In relation to the unchallenged solvent control, IL-1β-stimulationenhanced cytokine transcription levels of TNF-α, IL-1β, IL-8 and IL-6 already 4200-, 500-, 4700- and 1600-fold, respectively.

### 2.5. Determination of Cellular Protein Content and Metabolic Activity

To determine effects on the cellular protein content, the metabolic viability, and to preclude cytotoxic effects for qRT-PCR experiments, the sulforhodamine B (SRB) assay according to Skehan, et al. [36] and the alamarBlue^®^ assay were performed in HT-29 and HCEC-1CT cells. HT-29 (5500 cells/well) and HCEC-1CT cells (2000 cells/well) were seeded into 96-well plates and allowed to grow for 48 h. Cells were incubated for 5 h, including preincubation for 2 h with the substances (DON, NX-3, AURO and the respective combinations) followed by 3 h IL-1β co-treatment (25 ng/mL). Following 4 h of incubation, 10 µL alamarBlue^®^ reagent were added to the incubation solution. After 75 min, 70 µL of the supernatant were transferred into a black 96-well plate and fluorescence intensity was measured at 530/560 nm (excitation/emission) using a Synergy^TM^ H1 hybrid multimode reader (BioTek, Bad Friedrichshall, Germany). Subsequent to the fluorescence readout, cells were rinsed once with prewarmed PBS, fixed by the addition of 5% (*v/v*) trichloroacetic acid and incubated at 4 °C for 30 min. After the fixation, plates were washed four times with water, dried overnight at room temperature and stained for 1 h by adding a solution of 0.4% (*w/v*) SRB in 1% (*v/v*) acetic acid. Plates were washed twice with water and 1% acetic acid solution and dried at room temperature in the dark. Finally, 10 mM Tris buffer (pH 10; 100 µL/well) was added to dissolve the dye, and single wavelength absorbance (570 nm) was measured using a Synergy^TM^ H1 hybrid multimode reader (BioTek, Bad Friedrichshall, Germany). One percent (*v/v*) water (LC–MS grade) and 1% (*v/v*) DMSO with and without IL-1β stimulation served as solvent control, whereas 1% (*v/v*) triton X-100 was used as positive control. Cell-free blank values were subtracted and measured data were referred to the respective solvent control. Each cell line was tested in duplicate with a minimum of five independent experiments.

### 2.6. Statistical Analysis

Normal distribution of data was tested with the Shapiro–Wilk test. Correction of outliers was performed according to Nalimov. Statistical significances were calculated using OriginPro 2018G (Origin Lab, Northampton, MA, USA) applying one-way ANOVA followed by Bonferroni post hoc testing or one- and two-sample Student’s *t*-test.

## 3. Results

### 3.1. Impact of Fusarium Secondary Metabolites on LPS-Induced NF-κB Activation

Modulatory effects of DON and NX-3 and potential combinatory effects with the pro-oxidant co-contaminant AURO on LPS-induced NF-κB pathway activation were assessed in THP-1 NF-κB Luc Reporter Monocytes (Figure 2A). DON (1 µM), as well as NX-3, significantly increased the luciferase signal up to 241% ± 35% (DON) and 207% ± 29% of the LPS-induced signal (solid line, 100%). AURO caused an increase of luciferase signal limited to 139% ± 6% of the LPS-induced signal at 0.01 µM, whereas the other concentrations did not significantly modulate NF-κB activity. Combined incubations of AURO and DON or NX-3 resulted in a slightly, but not significantly reduced luminescence intensity compared to effects caused by DON and NX-3 alone.

Cell viability was monitored using the alamarBlue^®^ assay and all data were normalized to the evaluated metabolic activity (Figure 2B). A pronounced decrease of the fluorescence signal was determined in cells incubated with 5 and 10 μM DON and NX-3 and the respective combinations with 0.5 and 1 µM AURO, in line with a substantially decreased NF-κB activity. In most tested conditions, similar effects of NX-3 and DON could be observed. Only in the case of 5 µM NX-3 and the combination with 0.5 µM AURO, which showed similar effects on cell viability as the respective DON-treated samples, significant differences in NF-κB activity were determined.

### 3.2. Modulation of Cytokine Gene Transcription by Fusarium Secondary Metabolites

In order to assess the effects of DON and NX-3 (1 µM) and also their combination with AURO (0.1 µM) on NF-κB-dependent cytokine transcription, two colon cell lines, the cancer cells HT-29 and the noncancer cells HCEC-1CT, were exposed to the *Fusarium* secondary metabolites in the presence of the pro-inflammatory stimulus IL-1β (25 ng/mL). In HT-29 cells (Figure 3A), both trichothecene mycotoxins DON and NX-3 significantly increased TNF-α, IL-1β and IL-8 mRNA levels, whereas AURO did not lead to alterations of the analyzed cytokine transcription levels. Combinatory treatments of DON/AURO or NX-3/AURO resulted again in a slightly decreased signal when compared to the effects of the respective trichothecene single treatments.

In the noncancer cell line HCEC-1CT, a different cytokine pattern could be identified (Figure 3B). While in HT-29 the strongest induction was found for IL-1β transcription, followed by TNF-α and IL-8, in HCEC-1CT more TNF-α mRNA was present compared to the transcript levels of the other tested cytokines. In addition to IL-8 and IL-1β, IL-6 mRNA also could be identified in HCEC-1CT samples. While DON and NX-3 enhanced significantly the cytokine mRNA levels of all four target genes tested, respective cytokine transcripts were only marginally modulated after AURO treatment. Upon co-incubation of DON or NX-3 with AURO, increased mRNA levels similar to those following DON and NX-3 single substance treatment were identified.

When comparing the TNF-α transcription levels of the two colon cell lines, significant differences were observed (Figure 4A). While in the samples exposed to DON or NX-3 as single compounds no significant differences in TNF-α gene transcription were determined, combination with AURO decreased the TNF-α mRNA levels in HT-29 (DON/AURO: 1.6 ± 0.2 and NX-3/AURO: 1.3 ± 0.3 rel. transcription). These differences reached statistical significance in comparison to levels detected in HCEC-1CT cells. Combinations with AURO in the noncancer cell line resulted namely even in an increase in TNF-α gene transcription compared to the single compound treatments (DON/AURO: 2.7 ± 0.7 and NX-3/AURO: 2.7 ± 0.5 rel. transcription). Regarding IL-1β gene transcription (Figure 4B), significant differences between the two cell lines were already present in the samples exposed to DON and NX-3 as single compounds (DON: 3.1 ± 0.8 in HT-29 and 1.5 ± 0.2 in HCEC-1CT rel. transcription; NX-3: 2.8 ± 1.1 in HT-29 and 1.4 ± 0.3 in HCEC-1CT rel. transcription). Comparable to the effects in HT-29 cells, a slight decrease in IL-1β mRNA levels was observed, reaching statistical significance in the case of co-incubation with DON and AURO. IL-8 transcription levels did not differ between the two cell lines (Figure 4C).

### 3.3. Effects of Fusarium Secondary Metabolites on Cell Viability

In order to rule out cytotoxicity potentially compromising the analysis of immunomodulatory effects, the impact of the tested concentrations on cell viability was determined by the SRB (Figure 5A,B) and the alamarBlue^®^ assay (Figure 5C,D). In both cell lines, pronounced effects on the cellular protein content and on the metabolic activity after DON and NX-3 treatment for 5 h (last 3 h co-exposed to 25 ng/mL IL-1β) could be identified at concentrations ≥ 5 µM. AURO did not trigger any significant effects on cell viability except for the highest tested concentration (10 µM), which caused a pronounced decrease of the fluorescence signal in the alamarBlue^®^ assay in both cell lines. Partly significant differences between the samples treated in combination with AURO and the DON- or NX-3-single incubations were determined (Figure 5A–D, highlighted with ° symbols). However, due to the fact that the observed effects were of rather limited nature, no appropriate mathematical model for a correct evaluation of the combinatory interactions (e.g., the model of independent joint action [37] or the multiple drug effect equation [38]) could be applied.

## 4. Discussion

The novel type A trichothecene NX-3 was recently reported to possess comparable inhibitory potency on protein biosynthesis and similar cytotoxic potential as the well-known *Fusarium* mycotoxin DON [22,23]. In the present study, we explored in direct comparison to DON the immunomodulatory effects of NX-3, including its impact on NF-κB signaling pathway activation and on the expression of NF-κB target cytokines in two different intestinal cell lines, comparing the impact on a tumor cell line to noncancer cells. Furthermore, we investigated combinatory effects with AURO, a frequently co-occurring *Fusarium* secondary metabolite, which so far has not been assessed in any cell system regarding its immunomodulatory effects.

Activation of the NF-κB pathway plays a crucial role in inflammatory processes through its ability to induce the expression of various pro-inflammatory genes, including cytokines, chemokines, and adhesion molecules [39,40]. Numerous studies have reported a dysregulation of NF-κB signaling in patients suffering from irritable bowel syndrome [41,42,43] and identified this pathway as one of the major regulatory components in the complex pathogenesis and progression of chronic intestinal inflammatory disorders like Crohn’s disease and ulcerative colitis [44,45]. In order to assess the impact of the three *Fusarium* secondary metabolites DON, NX-3 and AURO on this important inflammatory signaling pathway, THP-1 monocytes carrying a NF-κB-inducible luciferase reporter construct were used. NX-3, as well as DON, activated the NF-κB pathway at a concentration of 1 µM (Figure 2A), whereas at higher concentrations, in line with a substantial decrease in cell viability (Figure 2B), a significantly reduced luciferase signal was observed. While this is the first report of NX-3-induced NF-κB activation, the respective effects of DON were already extensively studied during the last decades [12,46]. In Caco-2 cells, DON at concentrations between 1.6 and 16 µM slightly induced NF-κB pathway activity observed due to an increased phosphorylation of its inhibitor IκB and IL-8 secretion, whereas co-exposure to IL-1β or LPS resulted in a more pronounced pathway induction [16]. In HT-29 cells, microscopic localization of NF-κB p65 revealed a nuclear translocation within 15 min after DON treatment at a concentration of 0.8 µM, still active after 60 min [47]. Similar concentrations increased NF-κB p65 expression in HT-29 cells [48] and NF-κB binding in RAW 264.7 murine macrophage cells after 2 and 8 h of DON treatment in the presence and absence of LPS [49].

Due to the ability to influence the amount of intracellular reactive oxygen species (ROS), the transcription factor NF-κB and the regulation of downstream transcriptional targets play a crucial role in cell survival and in the prevention of cellular oxidative damage [50]. Low or transient levels of ROS are reported to trigger an inflammatory response through activation of the NF-κB signaling pathway [51,52,53]. Recently, the dimeric naphthoquinone AURO was shown to enhance intracellular ROS levels causing significant pro-oxidative DNA damage in HT-29 cells [25]. High levels of AURO contamination were reported in occurrence studies analyzing various food and feed components [4,5,29]. However, the impact of AURO on the NF-κB signaling pathway and potential combinatory interactions with co-occurring trichothecenes such as DON and NX-3 on the inflammatory response have not been addressed yet. AURO is known to be rather unstable and concentrations of about 10 µM were already reported to induce pronounced cytotoxic effects in the used cell systems [25,54]. We therefore used the low level of 1 µM to limit cytotoxicity and to be able to observe possible combinatory effects with the trichothecenes. Despite its reported pro-oxidative properties [25], AURO modulated NF-κB activity only marginally (Figure 2A). At 5 and 10 µM significant cytotoxic effects were detected, concomitantly with a decrease in NF-κB activity. Beside intracellular ROS formation, AURO was previously reported to enhance the ratio of GSSG/GSH and to induce significant oxidative DNA damage in HT-29 cells [25]. However, this pro-oxidative effect seems not sufficient to activate the NF-κB signaling cascade. Accordingly, no significant interactions were observed in the combinatory treatments of AURO with the trichothecenes DON and NX-3 (Figure 2A).

As a consequence of increased NF-κB transcription factor activity, enhanced expression levels of downstream target genes, including interleukins (IL-1β, IL-8 and IL-6) or the tumor necrosis factor (TNF-α) are expected [55]. Upregulation of cytokine mRNA expression can be triggered either transcriptionally or post-transcriptionally via increase of mRNA stability [56]. Since differences in cellular response between tumor and nontumor cells cannot be excluded, the impact of the tested mycotoxins on mRNA levels of pro-inflammatory cytokines were assessed using two intestinal epithelial cell models, the adenocarcinoma cell line HT-29 and the extended primary cell line HCEC-1CT (Figure 3A,B). In addition, co-treatments with IL-1β were used to mimic a potentially inflamed, pathologic IBD disordered intestinal epithelium [16,18,57]. In both cell lines, NX-3-enhanced mRNA levels of the assessed pro-inflammatory cytokines were comparable to DON treatment.

Several studies reported effects of DON on IL-secretion, focusing mainly on IL-8 [16,18], a cytokine acting as an early marker in inflammatory processes, mediating the activation and migration of neutrophils [58]. Six hours after DON exposure, Maresca, Yahi, Younes-Sakr, Boyron, Caporiccio and Fantini [18] reported a dose-dependent increase in IL-8 mRNA levels in Caco-2 cells at concentrations between 1 and 100 µM. Respective IL-8 protein levels were enhanced only at DON concentrations up to 25 µM, whereas higher concentrations resulted in a decrease of IL-8 secretion [18]. Similar effects on NF-kB-dependent IL-8 secretion were determined by Van De Walle, Romier, Larondelle and Schneider [16], revealing that IL-8 induction was potentiated upon pro-inflammatory stimulation by IL-1β and LPS. NX-3 was found to induce comparable effects as its type B trichothecene derivative DON, regarding cytotoxicity, induction of oxidative stress and GSH modulation [22,23]. In our study, both trichothecenes induced significantly enhanced transcript levels of different pro-inflammatory cytokines to a similar extent. Accordingly, a concomitant increase in cytokine secretion levels, as previously reported by Van De Walle, Romier, Larondelle and Schneider [16] and Maresca, Yahi, Younes-Sakr, Boyron, Caporiccio and Fantini [18], in Caco-2 cells after DON-treatment, is expected after DON as well as after NX-3 exposure in HT-29 and HCEC-1CT cells.

The effects observed on the NF-κB signaling pathway after AURO treatment indicate the conclusion that this *Fusarium* secondary metabolite lacks immunomodulatory potency, at least with respect to the spectrum of cytokines tested so far. These results argue for the fact that additional pro-inflammatory signaling pathways, such as the Toll-like receptors or retinoic acid-inducible gene-I-like receptors [59], are not substantially affected by low AURO concentration (0.1 µM, 5 h).

Studies focusing on the immunomodulatory effects of DON in normal, noncancer intestinal epithelial cells, to the best of our knowledge, are still limited to nontransformed porcine intestinal epithelial cells [60]. In that model, a concentration of 10 µM DON induced a pro-inflammatory response resulting in significantly increased transcription levels of mRNAs encoding for IL-8, IL-1α, IL-1β and TNF-α, reaching their maximum levels after 4 h of DON exposure. Similar results were reported using an ex vivo model of porcine jejunal explants [60].

Noteworthy, analysis of the cytokine mRNA levels in HT-29 and HCEC-1CT cells showed substantial differences in the cytokine transcription pattern (Figure 4). Under the applied experimental conditions, IL-6 transcript levels in HT-29 were below the detection limit, whereas in HCEC-1CT cells substantially higher IL-6 mRNA concentrations were observed (Figure 3). While in the case of IL-8 no significant differences between the two cell lines were noted, transcript analyses for TNF-α and IL-1β revealed significantly different expression patterns (Figure 4A,B). IL-1β transcription levels were much higher in HT-29 cells, compared to the levels in HCEC-1CT. Again, in both cell models, combinatory treatment with AURO did not substantially modulate IL-1β mRNA levels, compared to the effects caused by 1 µM DON or NX-3 alone. However, in the case of TNF-α significant differences between the two intestinal cell lines were determined only in combination with AURO. In HT-29 reduced amounts of TNF-α mRNA were found while HCEC-1CT cells showed increased transcript levels. As described in literature, HCEC-1CT cells are more susceptible to the toxic effects and stress induced by mycotoxins [22,25,61,62].

Since to-date no study evaluated the cytotoxic effects of NX-3 after short-term exposure and in order to rule out potential cytotoxicity affecting the analysis of cytokine transcripts by qRT-PCR, respective experiments were performed as part of this study (Figure 5A–D). A slight decrease in cell viability, more pronounced in HCEC-1CT cells, caused by the highest tested concentrations of DON and NX-3 could be determined. Varga, Wiesenberger, Woelflingseder, Twaruschek, Hametner, Vaclavikova, Malachova, Marko, Berthiller and Adam [22] reported NX-3 to induce pronounced cytotoxic effects in HT-29 and HCEC-1CT cells at concentrations ≥ 10 µM. Similar results were observed in the human hepatocyte carcinoma cell line HepG2 [23]. AURO affected cell viability only marginally at the highest concentration tested (10 µM). Jarolim, Wolters, Woelflingseder, Pahlke, Beisl, Puntscher, Braun, Sulyok, Warth and Marko [25] reported AURO to induce only minor cytotoxic effects in HT-29 and HCEC-1CT cells after 1 h of incubation at a concentration of 10 µM, whereas after 24 h, 5 µM AURO had already caused a statistically significant decrease in cell viability.

Taken together, despite the pro-oxidative properties of the potentially co-occurring bisnaphthoquinone derivative AURO, no immunomodulatory effects, neither alone nor in combination with NX-3 or DON were observed. The present study shows that the recently discovered type A trichothecene NX-3 can be seen as equipotent to DON in its potency to activate the NF-κB signaling pathway. Thereby, respective pro-inflammatory response was found not only in tumor cells but also in nontumorigenic intestinal cells. Altogether, this study underlines the importance to continuously explore the complex interaction between food contaminants and the intestinal inflammatory system. In order to allow proper risk assessment beyond healthy intestinal epithelia, known pathologic gastrointestinal tracts, e.g., from patients suffering IBDs that might be more sensitive to the effects of individual food contaminants and their mixtures, need to be taken into account.

## Figures and Tables

**Figure 1 microorganisms-08-00603-f001:**
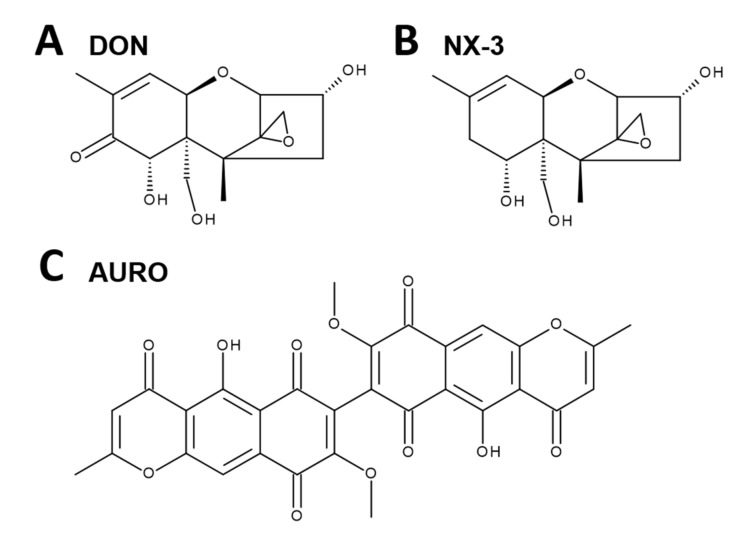
Chemical structures of the investigated *Fusarium* secondary metabolites: (**A**) deoxynivalenol (DON), (**B**) type A trichothecene (NX-3) and (**C**) aurofusarin (AURO).

**Figure 2 microorganisms-08-00603-f002:**
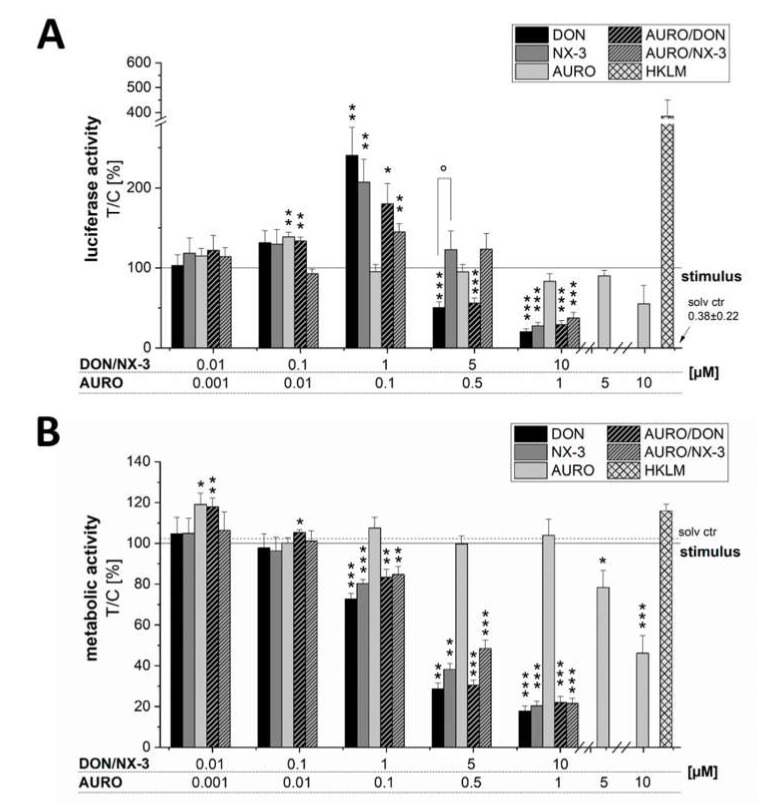
Activity of nuclear factor kappa B (NF-κB) in (**A**) lipopolysaccharide (LPS)-stimulated human monocytic THP1-Lucia™ NF κB cells. THP1-Lucia NF-κB cells were preincubated with the compounds (DON, NX-3, AURO and their combinations) for 2 h followed by an 18 h LPS challenge (10 ng/mL). Heat-killed *Listeria monocytogenes* (HKLM; 20 × 10^6^ cells/well) served as positive control for Toll-like receptor-mediated activation of the NF-κB pathway. Luminescence intensity data are expressed as mean values ± SE normalized to LPS-treated solvent control and to the respective cell viability data, assessed in (**B**) the alamarBlue^®^ cell viability assay of at least five independent experiments. One percent DMSO and 1% water (LC–MS grade) served as solvent control (dotted line). Significant differences to LPS, calculated with one-sample *t*-test, are indicated with * (*p* < 0.05), ** (*p* < 0.01) and *** (*p* < 0.001), whereas differences between DON and NX-3, calculated with a two-sample *t*-test, are indicated with ° (*p* < 0.05).

**Figure 3 microorganisms-08-00603-f003:**
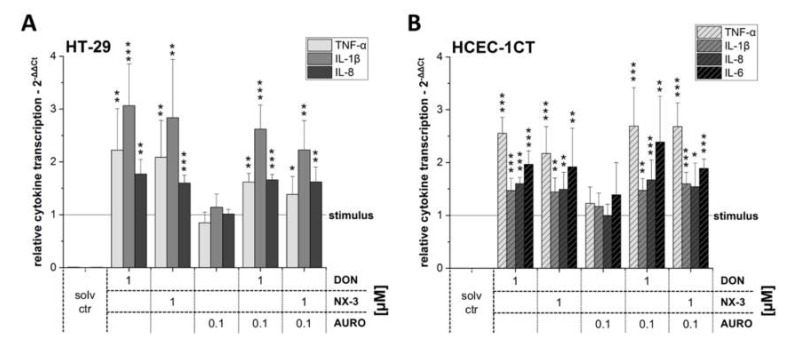
Relative gene transcription levels of TNF-α, IL-1β, IL-8 and IL-6 in (**A**) HT-29 and (**B**) HCEC-1CT cells (calibrator was IL-1β-treated solvent control, which was set to 1). Cells were preincubated with the compounds (DON, NX-3, AURO and their combinations) for 2 h followed by a 3 h IL-1β challenge (25 ng/mL). Relative transcript levels were measured with qRT-PCR. Data are expressed as mean values ± SD normalized to IL-1β-treated solvent control samples of at least five independent experiments. One percent DMSO and 1% water (LC–MS grade) served as solvent control. Significant differences to IL-1β-stimulation, calculated with two-sample *t*-test, are indicated with * (*p* < 0.05), ** (*p* < 0.01) and *** (*p* < 0.001).

**Figure 4 microorganisms-08-00603-f004:**
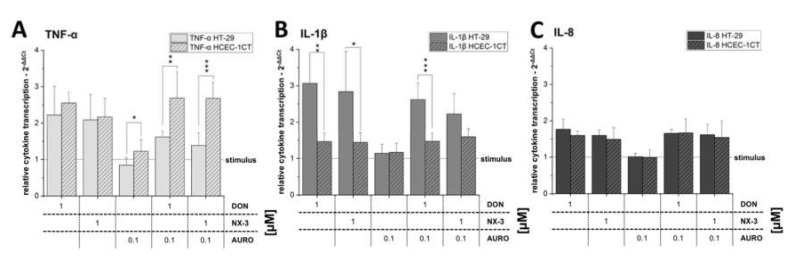
Relative gene transcription levels of (**A**) TNF-α, (**B**) IL-1β and (**C**) IL-8 in HT-29 and HCEC-1CT cells (calibrator was IL-1β-treated solvent control, which was set to 1). Cells were preincubated with the fungal metabolites (DON, NX-3, AURO and their combinations) for 2 h followed by a 3 h IL-1β challenge (25 ng/mL). Relative transcript levels were measured with qRT-PCR. Data are expressed as mean values ± SD normalized to IL-1β-treated solvent control samples of at least five independent experiments. One percent DMSO and 1% water (LC–MS grade) served as solvent control. Significant differences between the two cell lines, calculated with two-sample *t*-test, are indicated with * (*p* < 0.05), ** (*p* < 0.01) and *** (*p* < 0.001).

**Figure 5 microorganisms-08-00603-f005:**
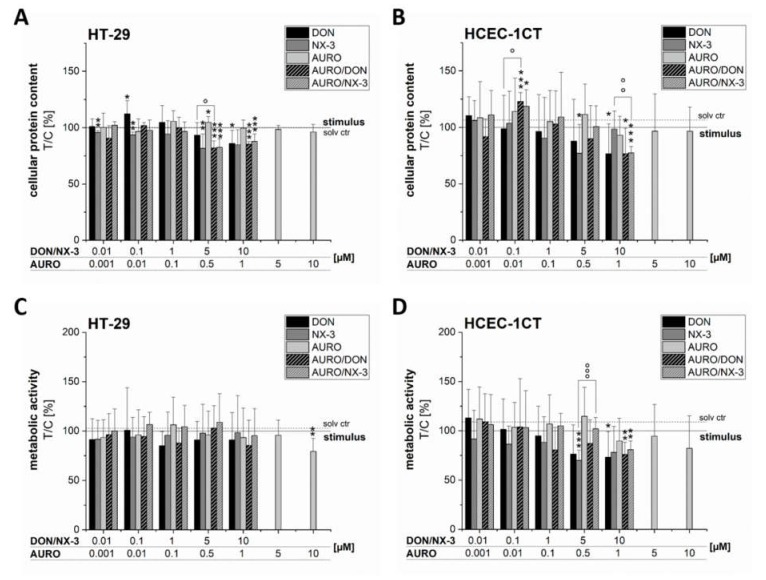
Effects of DON, NX-3, AURO and their combinations on the cellular protein content (**A,B**) and viability (**C,D**) of the two human colon cell lines HT-29 and HCEC-1CT determined in the sulforhodamine B (HT-29: **A**; HCEC-1CT: **B**) and alamarBlue^®^ assay (HT-29: **C**; HCEC-1CT: **D**). Cells were preincubated with the compounds (DON, NX-3, AURO and their combinations) for 2 h followed by a 3 h IL-1β challenge (25 ng/mL). Data are expressed as mean values ± SD normalized to IL-1β-treated solvent control samples of at least five independent experiments. One percent DMSO and 1% water (LC–MS grade) served as solvent control (dotted line). Significant differences to IL-1β-treated solvent control, calculated with one-sample *t*-test, are indicated with * (*p* < 0.05), ** (*p* < 0.01) and *** (*p* < 0.001), whereas differences between DON, NX-3 and their combinations with AURO, calculated with a two-sample *t*-test, are indicated with ° (*p* < 0.05), °° (*p* < 0.01) and °°° (*p* < 0.001).

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
