# Peer review of "Pro-Inflammatory Effects of NX-3 Toxin Are Comparable to Deoxynivalenol and not Modulated by the Co-Occurring Pro-Oxidant Aurofusarin"

_microorganisms, 2020, doi:10.3390/microorganisms8040603_

Round 1
Reviewer 1 Report
The authors present a very well written manuscript about the impact of NX-3 on the activity of the nuclear factor kappa B signaling pathway in direct comparison to DON, one of the most important mycotoxins in the area of food safety.
In my opinion, the manuscript is very well conducted and it represents an important step in the characterization of the new mycotoxins.
Author Response
Dear Reviewer 1,
we wish to thank you for the time you put in reviewing our paper. We are pleased and encouraged by your positive feedback.
Thank you for the revision.
Yours sincerely,
the authors
Reviewer 2 Report
In this manuscript, the toxicity of NX-3 was examined by culture cells. The subject of this manuscript is not microorganism but toxicology. Therefore, this manuscript is out of the scope of the journal “Microorganisms”. This manuscript should be submitted to other journals.
Before resubmitting this manuscript to other journals, check my comments.
1. Why did you use aurofusarin? Aurofusarin is not an important health hazard, and positive results were not obtained in this manuscript about autofusarin. For example, ochratoxin and fumonisin contaminate foods with DON. These mycotoxins should be tested.
2. In fig 3 and 4, the values of y-axis are difficult to understand. They should be expressed as “%” just like fig 2.
3. In fig 3 and 4, only the mRNA levels were quantified by RT-PCR. It is more important that the expression levels of the cytokine proteins were altered by DON or NX-3. The amount of proteins also should also be quantified.
Author Response
Dear Reviewer 2,
we wish to thank you for your constructive comments and the time you invested in the revision of our manuscript. Your comments regarding our submission provided valuable insights to refine its contents and analysis. In the following paragraphs, we try to address point-by-point the issues and suggestions you have raised in your feedback.
Reviewer 2: In this manuscript, the toxicity of NX-3 was examined by culture cells. The subject of this manuscript is not microorganism but toxicology. Therefore, this manuscript is out of the scope of the journal “Microorganisms”. This manuscript should be submitted to other journals.
We would like to thank you for your comment and agree that our study does not fit the general scope of the journal "Microorganisms". However, as our submission aimed for publication in the Special Issue "Fusarium: Mycotoxins, Taxonomy, Pathogenicity", focussing on the newest findings from the fields of plant pathology, genetics, chemistry, toxicology, and molecular biology of Fusarium research, we consider our manuscript to fulfill the requirements and to represent a well-suited and informative contribution to this Special Issue.
Reviewer 2: Why did you use aurofusarin? Aurofusarin is not an important health hazard, and positive results were not obtained in this manuscript about autofusarin. For example, ochratoxin and fumonisin contaminate foods with DON. These mycotoxins should be tested.
We totally agree with the reviewer, that the co-occurrence of prominent mycotoxins, such as DON, ochratoxin and fumonisins plays a major role with regard to food safety and risk assessment. Even though, no significant alterations of NF-κB pathway activity and only limited combinatory interactions with trichothecenes (DON, NX-3) were observed in this study, recently published data reported substantial aurofusarin-induced intestinal toxicities. Jarolim et al. (2018) observed cytotoxicity, oxidative stress, oxidative DNA damage and apoptosis induction in two different human intestinal cell lines following aurofusarin-exposure. Furthermore, Springler et al. (2016) reported significantly reduced TEER values, indicative for an impairment of intestinal barrier integrity, in differentiated intestinal porcine IPEC-J2 cells after 24, 48 and 72 h of aurofusarin exposure. Even though occurrence studies identified high concentrations of aurofusarin in Fusarium-contaminated food and feed samples (Streit et al. 2013; Uhlig et al. 2013), its immunomodulatory potential has not been assessed so far. Furthermore, as both, oxidative stress as well as a loss of intestinal barrier function pointed at a potential induction of intestinal inflammatory processes, we aimed to get a first insight into the inflammatory capacities of aurofusarin within this project.
Additionally, as co-occuring trichothecenes, such as DON and NX-3, have been previously reported to trigger oxidative stress (Mishra et al. 2014; Woelflingseder et al. 2018) we expected synergistic interactions of the prooxidant aurofusarin and these co-occuring trichothecene mycotoxins with regard to the activation of inflammatory signaling pathways.
Besides the characterization of aurofusarin, one of the main goals of this study was to elucidate the intestinal inflammatory potential of the novel type A trichothecene NX-3. Thus, further compound combinations would have unfortunately exceeded the capacities of this study. However, we agree with the reviewer that further toxicological assessement of revalent mycotoxin mixtures, including trichothecenes, ochratoxins and fumonisins is required to ensure proper risk assessment and food safety in future.
- Jarolim, K., Wolters, K., et al. (2018). The secondary Fusarium metabolite aurofusarin induces oxidative stress, cytotoxicity and genotoxicity in human colon cells. Toxicology letters, 284, 170-183.
- Mishra, S., Dwivedi, P D., et al. (2014). Role of oxidative stress in Deoxynivalenol induced toxicity. Food and Chemical Toxicology, 72, 20-29.
- Springler, A., Vrubel, G. J., et al. (2016). Effect of Fusarium-derived metabolites on the barrier integrity of differentiated intestinal porcine epithelial cells (IPEC-J2). Toxins, 8(11), 345.
- Streit, E., Schwab, C., et al. (2013). Multi-mycotoxin screening reveals the occurrence of 139 different secondary metabolites in feed and feed ingredients. Toxins, 5(3), 504-523.
- Uhlig, S., Eriksen, G. S., et al. (2013). Faces of a changing climate: Semi-quantitative multi-mycotoxin analysis of grain grown in exceptional climatic conditions in Norway. Toxins, 5(10), 1682-1697.
- Woelflingseder, L., Del Favero, G., et al. (2018). Impact of glutathione modulation on the toxicity of the Fusarium mycotoxins deoxynivalenol (DON), NX-3 and butenolide in human liver cells. Toxicology letters, 299, 104-117.
Reviewer 2: In fig 3 and 4, the values of y-axis are difficult to understand. They should be expressed as “%” just like fig 2.
The authors highly appreciate the reviewer's suggestion. Accordingly, in order to simplify the read-out of the data, the y-axes depicted in Figure 3 and Figure 4 have been adapted accordingly. As data have been normalized to transcription levels of the IL-1β-stimulated solvent control, we further indicated the relation between IL-1β-challenged and untreated solvent control in the figure legend and in detail in the Materials & Methods section.
Reviewer 2: In fig 3 and 4, only the mRNA levels were quantified by RT-PCR. It is more important that the expression levels of the cytokine proteins were altered by DON or NX-3. The amount of proteins should also be quantified.
We thank the reviewer for his suggestion to further improve this part of the manuscript. Due to the current COVID-19 crisis and respective restrictions by the Austrian government we are unfortunately not able to perform any experiment at the moment. However, we adapted parts of the discussion section of the paper to point out that previous toxicity studies, asessing the intestinal inflammatory potential of DON as single compound, have already measured transcript and protein levels of IL-8 in intestinal Caco-2 cells following DON-treatment (Maresca et al. 2008; Van De Walle et al. 2008). In parallel to significantly increased transcript levels of IL-8 (11-fold), approx. 10- and 40-fold enhanced IL-8 secretion levels were observed after 3 and 6 h of 10 µM DON-exposure, respectively (Maresca et al. 2008). Even though, the experimental set-up was different (cell-line, incubation time, IL-1β-stimulation) we cautiously conclude, that in our study protein levels rise similarly, according to the observed increase in transcript levels.
Furthermore, besides a significant increase in NF-κB pathway activity, Van De Walle et al. (2008) reported significantly enhanced IL-8 secretion levels after DON exposure (0.8 µM, 48 h) in Caco-2 cells. Il-1β-co-treatment further increased this effect (Van De Walle et al. 2008). Besides the results shown in Figure 2-4, previous studies have observed comparable toxic effects of NX-3 and DON. Due to the comparability of these two trichothecenes in the current study, we conclude that similar results can be expected with regard to cytokine secretion.
- Maresca, M., Yahi, N., et al. (2008). Both direct and indirect effects account for the pro-inflammatory activity of enteropathogenic mycotoxins on the human intestinal epithelium: Stimulation of interleukin-8 secretion, potentiation of interleukin-1β effect and increase in the transepithelial passage of commensal bacteria. Toxicology and applied pharmacology, 228(1), 84-92
- Van De Walle, J., Romier, B., et al. (2008). Influence of deoxynivalenol on NF-κB activation and IL-8 secretion in human intestinal Caco-2 cells. Toxicology letters, 177(3), 205-214.
Reviewer 3 Report
Comments on microorganisms 754985
Overall: This manuscript reports on the examination of the bioactivity of combinations of deoxynivalenol (DON), a novel DON analogue called NX-3 and a known, but untested Fusarium pigment aurofusarin. These three natural products, alone and in combination were tested for NF-kB activity as well as against a panel of cytokines. Additionally the authors tested the cytotoxicity of these combinations by determining the effect of exposure on cellular protein content.
Overall these experiments are well-executed and have been conducted in a manner consistent with standard practice. The results will be of interest to a broad audience as this is the first detailed examination of the bioactivity of aurofusarin against human cell lines. This is also the first detailed examination of the bioactivity of the novel DON analogue NX-3. This will be of interest to the natural products community, especially those interested in Fusarium toxins. This manuscript will give, for the first time, a detailed picture of the interaction of these molecules with respect to their bioactivity.
This manuscript should be published after the minor change noted below is incorporated.
Specific Comment:
Figure 2: In the bioassay data reported in Figure 2 there is a set of experiments referred to as “HKLM”. The meaning of this acronym should be clearly described in the Figure legend. This will prevent the reader from having to search through the Methods section to determine the meaning of this acronym.
Author Response
Dear Reviewer 3,
the authors would like to thank you for your precise revision and the constructive feedback regarding our mansucript. We highly appreciate your positive evaluation.
With regard to your comment on the figure caption of Figure 2, we would like to thank you for your suggestion. The figure caption was revised and a description of the acronym HKLM was inserted accordingly.
Once again, we would like to thank you for the time you put in the review of our paper.
Yours sincerely,
the authors
Round 2
Reviewer 2 Report
I have no more comments.